# Mediation of Healthy Behaviour on the Association of Frailty with Respiratory Diseases Mortality among 0.4 Million Participants: A Prospective Cohort Study from UK Biobank

**DOI:** 10.3390/nu14235046

**Published:** 2022-11-27

**Authors:** Min Du, Liyuan Tao, Min Liu, Jue Liu

**Affiliations:** 1Department of Epidemiology and Biostatistics, School of Public Health, Peking University, Beijing 100191, China; 2Research Center of Clinical Epidemiology, Peking University Third Hospital, Beijing 100191, China; 3Institute for Global Health and Development, Peking University, Beijing 100871, China; 4Global Center for Infectious Disease and Policy Research & Global Health and Infectious Diseases Group, Peking University, Beijing 100191, China; 5Key Laboratory of Reproductive Health, National Health and Family Planning Commission of the People’s Republic of China, Beijing 100191, China

**Keywords:** frailty, healthy behaviour, respiratory diseases, mortality, cohort study

## Abstract

The mutual relationship between frailty and healthy behaviour and its effect on respiratory diseases mortality remains largely unknown; this study aims to supplement related analysis on it by using a large sample cohort study. We included 411,987 participants from the UK Biobank study (2006–2021), and measured participants’ frailty phenotype and healthy behaviour index by using questionnaires and physical measurement. Mortality from respiratory diseases were obtained through linkage to registries. We used the cox proportional hazards model to explore the association of frailty with respiratory diseases mortality, and calculated the mediation proportion of the healthy behaviour. During a median follow-up of 12.48 years, and after adjustment for other covariates and healthy behaviour index, when compared with non-frail participants, being frail was associated with 2.68 times, 3.27 times, and 3.31 times higher risk of total respiratory diseases mortality, influenza and pneumonia mortality and chronic lower respiratory diseases mortality, respectively. The attenuated proportions mediated by healthy behaviour were 5.1% (95% CI: 4.4%, 5.9%), 3.0% (95% CI: 2.1%, 4.2%) and 6.0% (95% CI: 4.9%, 7.4%), respectively. Compared with non-frail individuals with four or five healthy behaviours, frail individuals with no or one healthy behaviour had higher risks of total respiratory diseases mortality (aHR = 4.59; 95% CI: 3.27, 6.45), influenza and pneumonia mortality (aHR = 4.55; 95% CI: 2.30, 9.03), as well as chronic lower respiratory diseases mortality (aHR = 12.70; 95% CI: 5.76, 27.96). Adherence to a healthy lifestyle therefore represents a potentially modifiable target for improving the harmful impact of frailty on reduced life expectancy as a result of respiratory diseases.

## 1. Introduction 

Respiratory diseases consistently rank among the top three fatal diseases in the world, and represent a large health burden across health systems [1]. The third leading cause of death in 2017 was chronic respiratory diseases, coming in just behind cardiovascular disease and neoplasms [2]. The World Health Organization (WHO) reported that over three million people died each year because of chronic obstructive pulmonary disease (COPD), accounting for nearly 6% of all deaths worldwide [3]. In the United Kingdom (UK), mortality from respiratory diseases was high between 1985 and 2015, especially for obstructive, interstitial, and infectious respiratory diseases [4]. Finding the influencing factors for reduced life expectancy due to respiratory diseases is therefore important for contributing to improve health and longevity.

Frailty is a clinical state, with an increased susceptibility to decompensation under physiological stress, and is associated with old age [5]. A large number of approaches have been used to identify frailty, of which the frailty phenotype as an operational definition of frailty now dominates scientific articles in this field [6]. Frailty is a risk factor in morbidity, mortality and disability [7]. A high prevalence of frailty has been reported in individuals with chronic respiratory diseases [8,9]. This may suggest that for the development and progression of respiratory diseases, frailty may be an independent risk factor. One recent study reported that the risk of hospital admission or death as a result of severe COVID-19 infection was higher among frail people [10]. However, studies on the relationship of frailty with deaths from respiratory diseases are very limited. Based on the high prevalence of frailty in respiratory disease patients, and a lack of evidence for the association between frailty and respiratory diseases mortality, this study aims to supplement that gap.

Healthy behaviour is commonly viewed as an important influencing factor for respiratory health, and may alleviate the health burden of respiratory diseases [11,12,13]. The healthy behaviour index, which consists of body mass index (BMI), smoking status, alcohol consumption, physical activity and diet, is a commonly used index used to explore the promotion and prevention of diseases or mortality [14,15]. Several studies reported that healthy behaviour might alleviate the poor effect of low social status on the mortality from cardiovascular diseases [15,16]. Healthy behaviour as an observed and modifiable factor, as well as its function in preventing the harmful effects of frailty on respiratory diseases mortality, is useful for providing behavioural guidance for patients at both clinical and public health levels. However, its association with frailty as a factor in respiratory diseases mortality remains unknown.

To provide further evidence of the relationship between healthy behaviour and frailty in the progression of respiratory diseases, the large sample size of the UK Biobank cohort data was used to evaluate the effect of frailty on different type of respiratory diseases mortality and the mediation function of healthy behaviour. After this, the interaction and joint associations between frailty and healthy behaviour with respiratory diseases mortality were analysed to supplement the limited amount of available research. Finally, a stratified analysis was carried out to explore the consistency of findings between sub-populations of different ages and gender. 

## 2. Materials and Methods

### 2.1. Study Population

For this study, data from the UK Biobank study (2006–2021) (data application number 79114, https://www.ukbiobank.ac.uk/ (accessed on 4 August 2022)), a large prospective cohort study of participants aged 37 to 73 years, was used for analysis. All participants were registered with the UK National Health Service (NHS) and resided within 40 km of one of 22 assessment centres across the United Kingdom (England, Wales, and Scotland). Details are described elsewhere [17,18]. The sample size of the UKB cohort data (2006–2021) was 502,414 (273,329 males and 229,085 females). Of these, and based on the cohort design, those with missing information in terms of frailty (n = 36,931, 7.35%), healthy behaviour index (n = 52,910, 10.35%) and other covariates (n = 586) were excluded. This yielded 411,987 participants who were included in the final analysis. Appendix A presents more detailed information on missing covariates. The National Information Governance Board for Health and Social Care, and the NHS North-West Multi-centre Research Ethics Committee approved the UK Biobank study. Electronically signed consent was provided by all participants. 

### 2.2. Assessment of the Frailty Phenotype 

Fried frailty phenotype, adapted for the UK Biobank study, used the following five criteria: weight loss, exhaustion, physical activity, walking speed and grip strength [6,10,19,20]. Participants provided baseline assessment data by means of touch-screen questionnaires and physical measurements (details can be seen in Appendix A).

A Jamar J00105 hydraulic hand dynamometer was used to measure the grip strength in whole kilogramme-force units. The participant was seated upright with their elbow by their side and flexed at 90°; they then kept their forearm facing forwards while resting on an armrest and exerted a single 3-s maximal grip effort. This was done individually in both the right and left arms to assess isometric grip force. The average values of the right and left forces were expressed in absolute units (kg) and used in subsequent analyses. Based on cut-offs stratified by gender and body mass index (BMI) from Fried et al.’s original description, a low grip strength could thus be assessed [6]. All components were summed up after coded as either zero or one. Of the three mutually exclusive groups, participants who fulfilled three or more criteria were classified as frail, those with one or two of the criteria were classified as pre-frail, and the others were considered to be robust [6,10,20]. Participants with missing data for any one criterion were excluded from this study.

### 2.3. Assessment of Healthy Behaviours and Other Covariates

A healthy behaviour index was constructed based on previous UK Biobank studies [14,15,21]. This was based on five factors, including body mass index (BMI) (kg/m^2^), smoking status, alcohol intake, physical activity and diet scores (details can be seen in Appendix A). Each behavioural component at a healthy level was assigned 1 point, and otherwise 0 point. As a result, the sum score of the healthy lifestyle points ranged between 0 and 5, which were further classified into three groups (no or one, two or three, four or five). The self-reported weekly and monthly intake of six classes of alcoholic drinks were used to assess alcohol intake. Moderate alcohol intake was defined as 5 to 15 g and 5 to 30 g of alcohol per day for women and men, respectively [15]. The international physical activity questionnaire was used to assess physical activity, based upon the total metabolic equivalent task (MET) minutes per week of all activities [22]. Sufficient physical activity was considered to be ≥735 MET min/week [15]. A touchscreen food frequency questionnaire, which included the six main food groups, was created to create diet scores (theoretical range: zero to seven); those with a score greater than or equal to four were considered to have a healthy diet [15,23].

Other covariates were obtained through questionnaires, including age, sex, race and ethnicity, educational level, Townsend deprivation index (TDI), general health, cancer, diabetes, cardiovascular disease, poor psychological health and family history of diseases (stroke, high blood pressure, chronic bronchitis/emphysema, Alzheimer’s disease/dementia, diabetes 2, Parkinson’s disease, severe depression, lung cancer, bowel cancer and breast cancer), sleep duration, and consumption of coffee and tea. As an indicator of socioeconomic status, the TDI indicates relative affluence as negative [24]. Sleep duration was categorised into three groups: short, normal, and long sleep durations, based on the National Sleep Foundation’s sleep time duration recommendations for adults and older adults [25]. 

### 2.4. Assessment of Respiratory Diseases Mortality

The NHS Information Centre (England and Wales) and the NHS Central Register (Scotland) provided the primary outcomes for this study. This included vital status, date of death, and underlying primary cause of death, as of 30 June 2020 [26]. Codes from the International Statistical Classification of Diseases and Related Health Problems, 10th Revision (ICD-10), were used to define specific causes of death as follows: J00–J99 total respiratory diseases; J09–J18 influenza and pneumonia; J40–J47 chronic lower respiratory diseases.

### 2.5. Statistical Analysis

Baseline characteristics were presented as mean (standard deviation, SD) or median (interquartile range, IQR) for continuous variables, and as a number (percentage) for categorical variables. Variance analysis for continuous variables and the χ^2^ test for categorical variables were used to test for differences between groups. 

A Cox proportional hazard regression was used to estimate the hazard ratios (HR) and a 95% confidence interval (CI) for the prospective association of outcomes associated with frailty or healthy behaviours. The proportional hazards assumption was tested using Schoenfeld residuals. Person years were calculated from baseline until the date of death from respiratory disease, or until end of follow-up, whichever occurred first. In terms of prior knowledge and descriptive statistics from the cohort, confounding was assessed by means of a DAG (directed acyclic graph) taken from the DAGitty v3.0 website (http://dagitty.net/dags.html# (accessed on 8 November 2022)) for this study (Appendix A) [13,16,27,28]. Model 1 included frailty (frail, pre-frail, robust), age (<65 years, ≥65 years), sex (male, female), TDI (continuous), educational level (college or university degree, A/AS levels or equivalent, O/GCSEs or equivalent, CSEs or equivalent, NVQ/HND/HNC equivalent, other professional qualifications), race and ethnicity (White, Black, Asian, mixed, other), general health (excellent, good, fair, poor), cancer (yes, no), diabetes (yes, no), cardiovascular disease (yes, no), poor psychological status (yes, no), family history (yes, no, unknown), sleep duration (normal, short, long), coffee intake (yes, no) and consumption of tea (continuous). Based on Model 1, Model 2 consisted of the same content plus the healthy behaviour index. The difference method was used to calculate the mediation proportion of healthy behaviour index on the association between frailty and death from respiratory diseases [16,29]. A stratified analysis was conducted, including healthy behaviour, in order to investigate associations of frailty with deaths from respiratory diseases. 

A product term of frailty (robust, frail) and healthy behaviours (no or one, four or five) was included to quantify the additive and multiplicative interactions in the model [16]. We measured the interaction of the product term on the multiplicative scale. The synergy index and corresponding 95% CI was used to measure the interaction on the additive scale [30]. In order to assess joint associations, nine groups were included, taking into account subject frailty (robust, pre-frail, frail) and healthy behaviour scores (no or one, two or three, four or five points), as well as estimated HR in the other eight groups compared with those who were robust and with four or five healthy lifestyle factors. The analyses were stratified by gender (men and women) and age group (<65 years and ≥65 years) in order to test the robustness and potential variations within different subgroups.

The following sensitivity analyses were used to assess the robustness of the results. Participants who had an outcome event during the first 5 years of follow-up were excluded. Multiple imputation via chained equations was used to impute all missing independent variables and test the influence of missing variables [16,28]. 

SAS 9.4 (SAS Campus Drive Cary, Cary, NC, USA) was used for performing the exact difference method. Other analyses were done using R software, version 4.2.1 (R Foundation). *p* values less than 0.05 (2-sided) were considered to be of statistical significance.

## 3. Results

### 3.1. Population Characteristics

Table 1 shows the baseline characteristics of all included participants. Among 411,987 participants (mean age 56.32 years, 53.7% men), 21,203 (5.1%) met the criteria for frailty. Frail adults tended to be older and from a lower social class. People who were men, non-white, in poorer general health, suffering from multiple comorbidities, with abnormal sleep durations and/or a lower coffee consumption were more likely to be found among the frail adult population.

### 3.2. Analysis of Healthy Behaviour on the Association of Frailty with Respiratory Diseases Mortality

During a median follow-up of 12.48 years (IQR, 11.56 to 13.29 years), 1646 deaths from respiratory diseases were recorded, of which, 393 and 722 were influenza and pneumonia deaths, and chronic lower respiratory diseases deaths, respectively. 

After being adjusted for healthy behaviour index and other covariates, and compared with non-frail participants, being pre-frail and frail were associated with 1.41 times (95% CI: 1.24, 1.60) and 2.68 times (95% CI: 2.26, 3.16) higher risk of total respiratory diseases mortality, respectively; 1.64 times (95% CI: 1.27, 2.12) and 3.27 times (95% CI: 2.30, 4.64) higher risk of influenza and pneumonia mortality, respectively; 1.52 times (95% CI: 1.23, 1.88) and 3.31 times (95% CI: 2.56, 4.26) higher risk of chronic lower respiratory diseases mortality, respectively (Table 2). The hazard ratios without adjustment for healthy behaviour index were larger. Sensitivity analyses showed that results were stable (Appendix A). The associations of each frailty index with respiratory diseases mortality can be seen in Appendix A. When frail individuals are compared with robust individuals, the proportion mediated by healthy behaviours was 5.1% (95% CI: 4.4% to 5.9%) for total respiratory diseases mortality, 3.0% (95% CI: 2.1% to 4.2%) for influenza and pneumonia mortality, and 6.0% (95% CI: 4.9% to 7.4%) for chronic lower respiratory diseases mortality (Table 2). Appendix A shows the results stratified by healthy behaviour index: being frail was associated with higher risks of total respiratory diseases mortality, influenza and pneumonia mortality among individuals of various healthy behaviour subgroups, while the associations were stronger among those with no or one healthy behaviour. 

A higher healthy behaviour index, compared with those with scores of zero or one, were associated with 25% to 72% lower risks of mortality (Appendix A). Adults with four or five healthy behaviours showed lower risks of total respiratory diseases mortality, as well as influenza and pneumonia mortality, while the associations were stronger among those deemed to be pre-frail (Figure 1). 

### 3.3. Interaction and Joint Analysis of Healthy Behaviour and Frailty with Respiratory Diseases Mortality

No significant multiplicative or additive interaction was found between healthy behaviour and frailty in terms of its effect on respiratory disease mortality (Table 2). Figure 2 shows the joint associations of healthy behaviour and frailty. Hazard ratios for frail individuals with scores of zero or one on the healthy behaviour index, compared with those who were not frail and scored four or five on the healthy behaviour index were 4.59 (95% CI: 3.27, 6.45) for total respiratory diseases mortality, 4.55 (95% CI: 2.30, 9.03) for influenza and pneumonia mortality and 12.70 (95% CI: 5.76, 27.96) for chronic lower respiratory diseases mortality. Results were stable throughout all sensitivity analyses (Appendix A).

### 3.4. Healthy Behaviour and Frailty with Respiratory Diseases Mortality among Different Sub-Populations

Appendix A show the results stratified by gender and age group. In contrast to chronic lower respiratory diseases, the associations of frailty with influenza and pneumonia mortality, and with total respiratory disease mortality were stronger in men compared to women, and were also stronger in younger compared to older adults (Appendix A). Except influenza and pneumonia mortality, the joint associations of healthy behaviour and frailty with chronic lower respiratory diseases and total respiratory disease mortality were stronger in men compared to women; in younger compared to older adults (Appendix A).

## 4. Discussion

To our knowledge, this was the first study to explore the association between frailty and respiratory diseases mortality, and the first to examine whether and to what extent healthy behaviour mediates the association of frailty with respiratory diseases mortality. In this prospective analysis of more than 400,000 participants from the UK Biobank study, it was found that frailty was associated with higher risks of mortality from total respiratory diseases, influenza and pneumonia, and from chronic lower respiratory diseases, and that 3.0% to 6.0% of the associations were mediated by healthy behaviour. In addition, the highest risks of respiratory diseases mortality were seen in frail adults with no or one healthy behaviour.

Frailty as a factor in mortality from respiratory diseases is rarely discussed, though it is considered to be a factor in mortality from all diseases (all-cause mortality). A large cohort study consisting of 512,723 participants from China found that each 0.1 increment in the frailty index was associated with a 68% higher risk of all-cause mortality [31]. Hanlon et al. reported that all-cause mortality was associated with frailty and pre-frailty, after adjusting for multi-morbidity, socioeconomic status, body-mass index, smoking and alcohol use, when using the UK Biobank data [19]. To date, limited studies reported the associations of frailty with respiratory diseases mortality. One study by the China Kadoorie Biobank reported that for each 0.1 increment in the frailty index, the corresponding HR for risk of respiratory disease mortality was 2.54 (95% CI: 2.45, 2.63) [31]. Recently, Petermann-Rocha et al. found that frailty was associated with a higher risk of hospital admissions or death due to severe COVID-19 infection [10]. The current study comprehensively analysed the associations between frailty and different types of respiratory disease mortality. It was found that frailty could lead to a 2.68, 3.27 and 3.31-fold increase in total respiratory diseases mortality, influenza and pneumonia mortality and chronic lower respiratory diseases mortality, respectively. The association between frailty and specific types of respiratory disease mortality suggests that frailty may be a robust surrogate measure for predicting the risk of death from respiratory diseases.

The expanded findings showed definite age and gender differences for the association of frailty with respiratory diseases mortality. In contrast to chronic lower respiratory diseases, the associations of frailty with influenza and pneumonia mortality, as well as with total respiratory diseases mortality, were stronger in men compared to women, and also stronger in younger compared to older adults. The above findings suggest that more research into frailty with respiratory diseases mortality, along with the development of screening and intervention programmes are warranted, especially in younger male populations. 

Our study found that in UK adults up to 6.0% of the association between frailty and respiratory diseases mortality was explained by healthy behaviours. A healthy lifestyle is an important influencing factor for health, and previous studies have reported that it may also alleviate the risk of death [11,12,13]. Zhu et al. reported that participants with five healthy behaviours was 0.26 (95% CI: 0.14, 0.48) for respiratory diseases mortality, compared with those without healthy factors [11]. The current study also found that increased healthy behaviours led to a lower risk of respiratory diseases mortality. One study reported that healthy behaviour could alleviate the poor effect of low social status on mortality from cardiovascular disease [15,16]. The joint association of healthy behaviour and frailty was also observed. Except for mortality associated with influenza and pneumonia, joint associations of healthy behaviour and frailty on mortality from all respiratory diseases and chronic lower respiratory diseases were stronger in men compared to women. Joint associations of healthy behaviour and frailty on mortality from respiratory diseases were all stronger in younger compared to older adults. This study first explored the idea that healthy behaviour could decrease the risk of frailty on respiratory disease mortality. Although the mediation proportion was slightly lower than 10%, it still indicated that moderate reductions in the effect of frailty on respiratory disease mortality could be achieved through promoting a healthier lifestyle.

The major strength of this study was the large sample size which allowed our results to have sufficient statistical power. In addition, the UK Biobank data used information on deaths from the NHS Information Centre and the NHS Central Register to ensure sufficient accuracy. The study does also have its limitations though. First of all, certain self-reported information may be susceptible to recall bias. As a result, further objective measurements are urgently needed in this field. In addition, random controlled trials are required in future to avoid the existing potential residual, in spite of the estimation models used being adjusted for a wide range of known confounding factors.

## 5. Conclusions

In conclusion, based on a large nationwide cohort from the UK, this prospective analysis found that frailty was associated with respiratory diseases mortality, while healthy behaviour was able to mediate this frailty to a certain extent. Therefore, healthy lifestyle promotion may be able to substantially attenuate the contribution of frailty to respiratory diseases mortality. The results obtained highlighted the risk of the progression of respiratory disease in frailty. Adherence to healthy behaviour represented potentially modifiable targets for improving the harmful impact of frailty on life expectancy from respiratory diseases. In the future, other approaches, such as random controlled trials, are required to explore the benefits of healthy behaviour in reducing the burden of respiratory disease. These novel findings may provide relevant evidence for clinical and public health policies on respiratory disease management.

## Figures and Tables

**Figure 1 nutrients-14-05046-f001:**
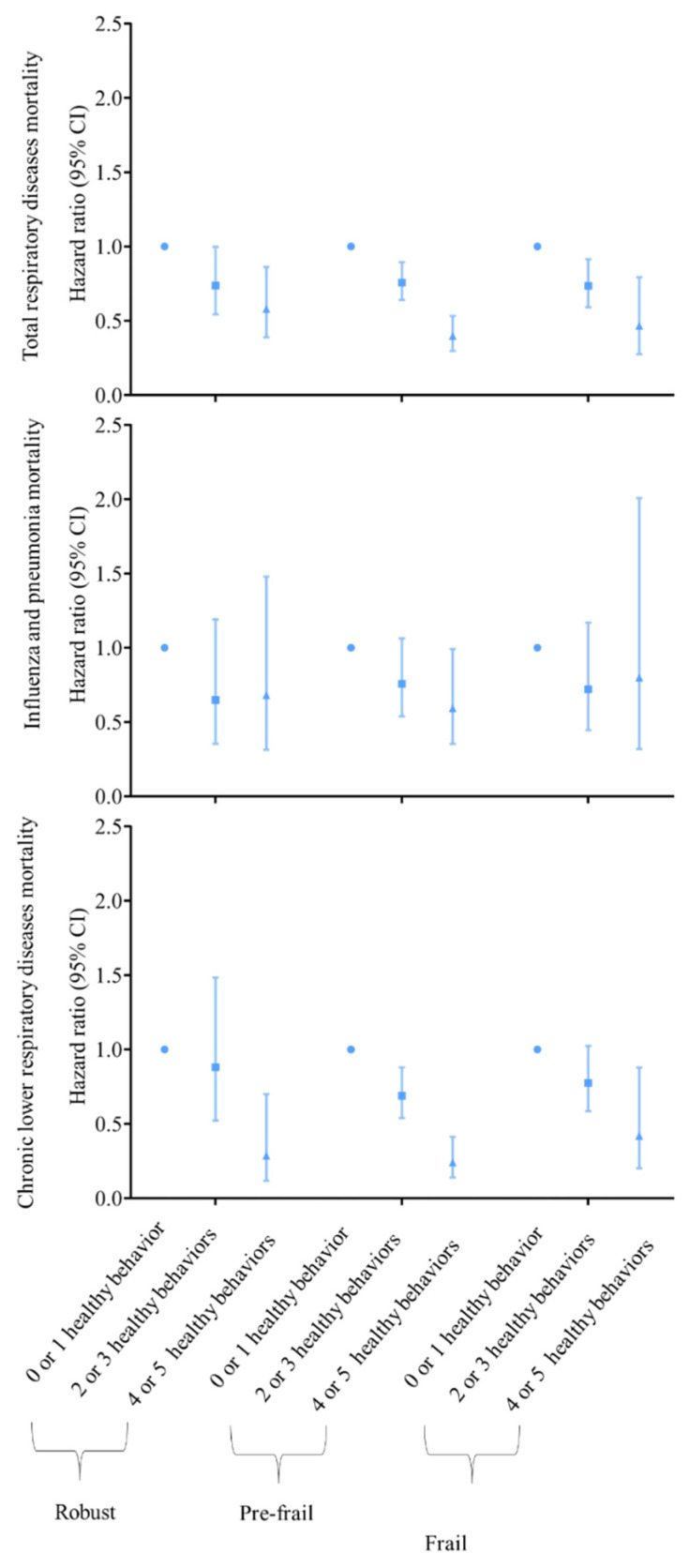
Associations of healthy behaviour with respiratory diseases mortality by frailty. Models all adjusted for age, sex, TDI, educational level, race and ethnicity, general health, cancer, diabetes, cardiovascular disease, poor psychological status, family history, sleep duration, coffee intake and consumption of tea. HR = hazard ratio. 95% CI = 95% confidence interval.

**Figure 2 nutrients-14-05046-f002:**
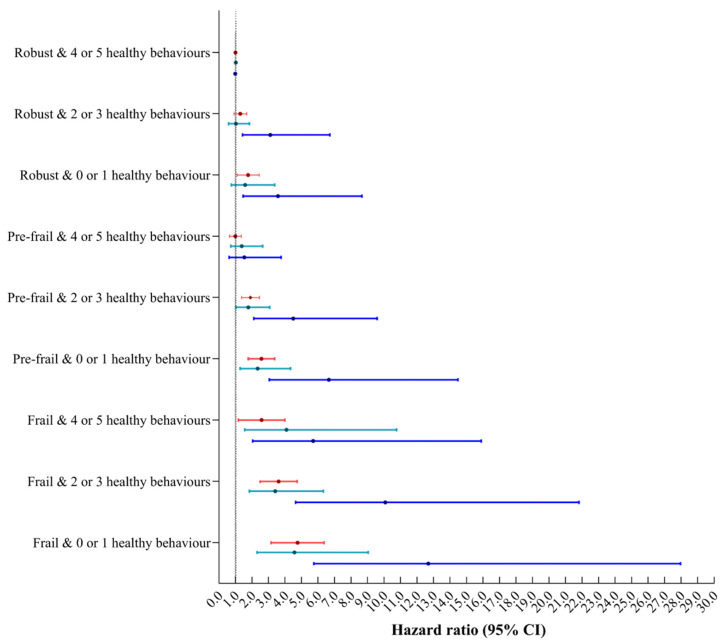
Joint associations of healthy behaviour and frailty with respiratory diseases mortality. Models all adjusted for age, sex, TDI, educational level, race and ethnicity, general health, cancer, diabetes, cardiovascular disease, poor psychological status, family history, sleep duration, coffee intake and consumption of tea. HR = hazard ratio. 95% CI = 95% confidence interval. Red showed total respiratory diseases mortality, green showed influenza and pneumonia mortality, blue showed chronic lower respiratory diseases mortality.

**Table 1 nutrients-14-05046-t001:** Baseline characteristics of included participants.

Characteristics	N (%/SD/IQR)	Robust (n, %)	Pre-Frail (n, %)	Frail (n, %) *
	411,987	161,787	228,997	21,203
Mean age (SD), year	56.32 (8.06)	56.38 (8.04)	56.17 (8.10)	57.58 (7.72)
Age, year				
<65	337,623 (81.9)	132,607 (82.0)	188,449 (82.3)	16,567 (78.1)
≥65	74,364 (18.1)	29,180 (18.0)	40,548 (17.7)	4636 (21.9)
Sex				
Male	221,227 (53.7)	80,441 (49.7)	127,373 (55.6)	13,413 (63.3)
Female	190,760 (46.3)	81,346 (50.3)	101,624 (44.4)	7790 (36.7)
Townsend deprivation index (median [IQR])	−2.28 [−3.71, 0.22]	−2.50 [−3.83, −0.26]	−2.20 [−3.67, 0.36]	−0.95 [−3.05, 2.23]
Race and ethnicity				
White	394,334 (95.7)	156,469 (96.7)	218,349 (95.4)	19,516 (92.0)
Black	5178 (1.3)	1710 (1.1)	3028 (1.3)	440 (2.1)
Asian	7234 (1.8)	1906 (1.2)	4530 (2.0)	798 (3.8)
Mixed	2277 (0.6)	798 (0.5)	1327 (0.6)	152 (0.7)
other	2964 (0.7)	904 (0.6)	1763 (0.8)	297 (1.4)
Educational level				
College or University degree	144,367 (35.0)	58,895 (36.4)	80,474 (35.1)	4998 (23.6)
A/AS levels or equivalent	48,549 (11.8)	18,874 (11.7)	27,474 (12.0)	2201 (10.4)
O/GCSEs or equivalent	88,970 (21.6)	34,559 (21.4)	49,860 (21.8)	4551 (21.5)
CSEs or equivalent	21,460 (5.2)	8303 (5.1)	11,929 (5.2)	1228 (5.8)
NVQ/HND/HNC equivalent	26,656 (6.5)	11,109 (6.9)	14,140 (6.2)	1407 (6.6)
Other professional qualifications	81,985 (19.9)	30,047 (18.6)	45,120 (19.7)	6818 (32.2)
General health				
Excellent	73,393 (17.8)	36,053 (22.3)	36,594 (16.0)	746 (3.5)
Good	242,870 (59.0)	102,053 (63.1)	134,438 (58.7)	6379 (30.1)
Fair	80,381 (19.5)	22,304 (13.8)	49,927 (21.8)	8150 (38.4)
Poor	15,343 (3.7)	1377 (0.9)	8038 (3.5)	5928 (28.0)
Cancer				
No	380,589 (92.4)	150,267 (92.9)	211,477 (92.3)	18,845 (88.9)
Yes	31,398 (7.6)	11,520 (7.1)	17,520 (7.7)	2358 (11.1)
Diabetes				
No	392,321 (95.2)	157,252 (97.2)	217,037 (94.8)	18,032 (85.0)
Yes	19,666 (4.8)	4535 (2.8)	11,960 (5.2)	3171 (15.0)
Poor psychological status				
No	271,104 (65.8)	114,638 (70.9)	146,557 (64.0)	9909 (46.7)
Yes	140,883 (34.2)	47,149 (29.1)	82,440 (36.0)	11,294 (53.3)
Cardiovascular disease				
No	294,071 (71.4)	121,961 (75.4)	161,137 (70.4)	10,973 (51.8)
Yes	117,916 (28.6)	39,826 (24.6)	67,860 (29.6)	10,230 (48.2)
Family history				
No	32,504 (7.9)	13,537 (8.4)	17,745 (7.7)	1222 (5.8)
Yes	370,268 (89.9)	144,927 (89.6)	206,038 (90.0)	19,303 (91.0)
Unknown	9215 (2.2)	3323 (2.1)	5214 (2.3)	678 (3.2)
Sleep duration				
Normal	300,613 (73.0)	123,540 (76.4)	165,231 (72.2)	11,842 (55.9)
Short	98,229 (23.8)	34,277 (21.2)	56,490 (24.7)	7462 (35.2)
Long	13,145 (3.2)	3970 (2.5)	7276 (3.2)	1899 (9.0)
Tea intake (median [IQR]), cups/day	3.00 [1.00, 5.00]	3.00 [1.00, 5.00]	3.00 [1.00, 5.00]	3.00 [1.00, 5.00]
Coffee intake				
No	88,392 (21.5)	32,851 (20.3)	49,563 (21.6)	5978 (28.2)
Yes	323,595 (78.5)	128,936 (79.7)	179,434 (78.4)	15,225 (71.8)

* All *p* values < 0.0001.

**Table 2 nutrients-14-05046-t002:** Associations of frailty with respiratory diseases mortality.

	Hazard Ratio (95% CI)		Mediation Proportion (%) (95% CI) ^†^
	Unadjusted for Healthy Behaviour Index	Adjusted for Healthy Behaviour Index *	
Total respiratory diseases ^‡^			
Robust	1 (Reference)	1 (Reference)	
Pre-frail	1.44 (1.27, 1.63)	1.41 (1.24, 1.60)	5.1 (3.3, 7.9)
Frail	2.83 (2.39, 3.34)	2.68 (2.26, 3.16)	5.1 (4.4, 5.9)
Influenza and pneumonia ^§^			
Robust	1 (Reference)	1 (Reference)	
Pre-frail	1.66 (1.28, 2.14)	1.64 (1.27, 2.12)	2.4 (0.9, 6.1)
Frail	3.40 (2.40, 4.81)	3.27 (2.30, 4.64)	3.0 (2.1, 4.2)
Chronic lower respiratory diseases ^||^			
Robust	1 (Reference)	1 (Reference)	
Pre-frail	1.57 (1.27, 1.94)	1.52 (1.23, 1.88)	6.0 (3.4, 10.4)
Frail	3.55 (2.76, 4.58)	3.31 (2.56, 4.26)	6.0 (4.9, 7.4)

Models all adjusted for age, sex, TDI, educational level, race and ethnicity, general health, cancer, diabetes, cardiovascular disease, poor psychological status, family history, sleep duration, coffee intake and consumption of tea. * Multiplicative interaction was evaluated using hazard ratios for the product term between the healthy behaviours (no or one v four or five) and frailty (robust v frail), and the multiplicative interaction was statistically significant when its confidence interval did not include 1. Additive interaction was evaluated using the synergy index between the healthy behaviours (no or one v four or five) and frailty (robust v frail), and the additive interaction was statistically significant when its confidence interval did not include 1 [16,30]. ^†^ All *p* < 0.05. **^‡^** Multiplicative interaction: 0.84 (95% CI: 0.43, 1.61), *p* = 0.594; additive interaction: the synergy index = 0.24 (95% CI: 0.04, 1.45). **^§^** Multiplicative interaction: 1.35 (95% CI: 0.41, 4.43), *p* = 0.618; additive interaction: the synergy index = 0.84 (95% CI: 0.20, 3.53). **^||^** Multiplicative interaction: 1.37 (95% CI: 0.43, 4.35), *p* = 0.593; additive interaction: the synergy index = 0.17 (95% CI: 0.01, 3.15). HR = hazard ratio. 95% CI = 95% confidence interval.

## Data Availability

The UK Biobank datasets are openly available by submitting a data request proposal from https://www.ukbiobank.ac.uk/ (accessed on 9 June 2022). We are authorised to access the database through the Access Management System (AMS) (Application number: 79114).

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
