# Peer review of "Mediation of Healthy Behaviour on the Association of Frailty with Respiratory Diseases Mortality among 0.4 Million Participants: A Prospective Cohort Study from UK Biobank"

_nutrients, 2022, doi:10.3390/nu14235046_

Round 1
Reviewer 1 Report
This study is statistically strong and rather well described with a very large N.
Some additional information should be provided in the method section to allow a better understanding of the study by the reader.
My comments are included in the PDF

Author Response
Reviewer(s)' Comments to Author:
Reviewer: 1
This study is statistically strong and rather well described with a very large N.
Response: Thanks.
Some additional information should be provided in the method section to allow a better understanding of the study by the reader.
Response: Thanks. We have revised this manuscript based on the review report from review 1. Specific comments as follows:
--title: among 0.4 million participants is not useful, it is more useful to know in which country the study took place
Response: Thanks. We have changed title “Mediation of healthy behaviours on the association of frailty with respiratory diseases mortality among 0.4 million partici-pants: a prospective cohort study” as “Mediation of healthy behaviours on the association of frailty with respiratory diseases mortality among 0.4 million participants: a prospective cohort study from UK Biobank”
--abstract: The purpose of the study is not clearly defined. I think it is necessary to add one more sentence to explain how these data are obtained. the 95% CIs must be added to the aHR.
Response: Thanks. We have restated our purpose of this study as shown in page line : “Little is known about the mutual relationship of frailty and healthy behaviours on respiratory diseases mortality.” changed as “The mutual relationship of frailty and healthy behaviours on respiratory diseases mortality was still unknown, so we aimed to supplement related analysis on it by using a large sample cohort study.”.
We have added related information on obtained data as shown in page 1 line 17-20: “We included 411987 participants from UK Biobank (2006-2021), and measured participants’ frailty phenotype and healthy behaviours index.” changed as “We included 411987 participants from the UK Biobank study (2006-2021), and measured participants’ frailty phenotype and healthy behaviours index by using questionnaires and physical measurement. Mortality from respiratory diseases were obtained through linkage to registries.”.
We have added 95% CIs to the aHR as shown in page 1 line 29-31: “Compared with non-frail individuals with four or five healthy behaviours, frail individuals with no or one healthy behaviours had higher risks of total respiratory diseases mortality (aHR=4.59), influenza and pneumonia mortality (aHR=4.55), and chronic lower respiratory diseases mortality (aHR=12.70).” changed as “Compared with non-frail individuals with four or five healthy behaviours, frail individuals with no or one healthy behaviours had higher risks of total respiratory diseases mortality (aHR=4.59; 95% CI: 3.27, 6.45), influenza and pneumonia mortality (aHR=4.55; 95% CI: 2.30, 9.03), ands well as chronic lower respiratory diseases mortality (aHR=12.70; 95% CI: 5.76, 27.96)”.
--introduction: I find this background part a bit short, it should be developed a bit so that we understand the context, the issue and the interest of the study
Response: Thanks. We have developed introduction to make the context, the issue and the interest of the study clearer as shown in page 2 line 46-47, 57-60 and 68-72.
-- Materials and Methods: i would add in this method section a sample size section, which explains that there is no sample size calculation, but a pragmatic choice of sample size, so that all the persons who met the inclusion criteria of this primary cohort study were included. it is necessary to explain what this application
Response: Thanks. We have added information to explain here is no sample size calculation as shown in page 2 line 89-93 .: “The sample size of the UKB cohort data (2006-2021) was 502,414 (273,329 males and 229,085 females). Of these, and based on the cohort design, those with missing infor-mation in terms of frailty (n=36,931, 7.35%), healthy behaviour index (n=52,910, 10.35%), and other covariates (n=586) were excluded. This yielded 411,987 participants who were included in the final analysis.”.
We have explained the number of this application as shown in page line: “application 79114” changed as “data application number 79114”.
Reviewer 2 Report
The article is interesting because it studies how frailty can influence respiratory diseases. The design is robust, a prospective cohort study that significantly strengthens the study.
An important issue is the high number of variables for which it is adjusted. This could lead to bias. On the other hand, studying the associations between variables can be difficult. For all these reasons, a figure with a DAG (dag-directed acyclic graph) showing the relationship between the study's variables should be included in the text. DAGs allow drawing a theoretical map of the relationships among variables determining what variables should be controlled in a multivariate model, and avoiding bias Authors can find an example of its use at https://www.mdpi.com/2077-0383/10/24/5854/htm
Authors can draw the DAG easily using the free program Daggity, Available at http://dagitty.net/
Try to use the same verb tense within a paragraph. There are paragraphs in which one sentence is in the present tense and another in the past tense.
In the section on material and methods, provide a link to the Biobank, for example https://www.ukbiobank.ac.uk/
In the body of the article, references to supplementary tables and figures should be made correctly. Now only Table S2 is referenced. In the body of the article, reference is made to tables in the appendix and are cited, for example in lines 214- 215, "More healthy behaviours, compared with no or one healthy behaviour, were associated with 25% to 72% lower risks of mortality (Appendix Table 5)." This generates much confusion because in the appendix, the numbering of the table is not Table 5 but Table S5). It should be cited as table S5. This error must be corrected so as not to hinder the reading of the article.
Please rewrite the phrase "Except difference method was done using SAS 9.4 version," as it can confuse. I suggest the following wording:
SAS 9.4 was used for performing the exact difference method. Other analyses were done using R software, version 4.2.1.
Please review lines 287-288. Not understood "Healthy behaviours as important influencing factors for health," It should probably be "are" Instead of "as"
Author Response
Reviewer: 2
The article is interesting because it studies how frailty can influence respiratory diseases. The design is robust, a prospective cohort study that significantly strengthens the study.
Response: Thanks.
An important issue is the high number of variables for which it is adjusted. This could lead to bias. On the other hand, studying the associations between variables can be difficult. For all these reasons, a figure with a DAG (dag-directed acyclic graph) showing the relationship between the study's variables should be included in the text. DAGs allow drawing a theoretical map of the relationships among variables determining what variables should be controlled in a multivariate model, and avoiding bias Authors can find an example of its use at https://www.mdpi.com/2077-0383/10/24/5854/htm. Authors can draw the DAG easily using the free program Daggity, Available at http://dagitty.net/
Response: Thanks for your good suggestions. We mainly focus on the associations between frailty and respiratory diseases mortality, and the mediation function of health behaviours, so we adjusted these covariates based on previous studies (Inoue-Choi M, Ramirez Y, Cornelis MC, Berrington de González A, Freedman ND, Loftfield E. Tea Consumption and All-Cause and Cause-Specific Mortality in the UK Biobank: A Prospective Cohort Study. Ann Intern Med. 2022;175(9):1201-1211. doi:10.7326/M22-0041; Liu D, Li ZH, Shen D, et al. Association of Sugar-Sweetened, Artificially Sweetened, and Unsweetened Coffee Consumption With All-Cause and Cause-Specific Mortality: A Large Prospective Cohort Study. Ann Intern Med. 2022;175(7):909-917. doi:10.7326/M21-2977; Zhang YB, Chen C, Pan XF, et al. Associations of healthy lifestyle and socioeconomic status with mortality and incident cardiovascular disease: two prospective cohort studies. BMJ. 2021;373:n604. Published 2021 Apr 14. doi:10.1136/bmj.n604). We have added a DAG to make the relationship between variables clearly as shown in page 4 line 163-166 and Figure S1: “In terms of prior knowledge and descriptive statistics from the cohort, confounding was assessed by means of a DAG (directed acyclic graph) taken from the DAGitty v3.0 website (http://dagitty.net/dags.html#) for this study (Figure S1) [13,16,27,28].”.
Try to use the same verb tense within a paragraph. There are paragraphs in which one sentence is in the present tense and another in the past tense.
Response: Thanks. We have used use the same verb tense within a paragraph and corrected language issues of this article. Besides, all the manuscript has been copy edited for proper English language at LetPub which is an author service brand owned and operated by Accdon LLC. The language editors are native English speakers with long-term experience in editing scientific and technical manuscripts.
In the section on material and methods, provide a link to the Biobank, for example https://www.ukbiobank.ac.uk/
Response: Thanks. We have added a link to the Biobank. In addition, Data Availability Statement also provided this link.
In the body of the article, references to supplementary tables and figures should be made correctly. Now only Table S2 is referenced. In the body of the article, reference is made to tables in the appendix and are cited, for example in lines 214- 215, "More healthy behaviours, compared with no or one healthy behaviour, were associated with 25% to 72% lower risks of mortality (Appendix Table 5)." This generates much confusion because in the appendix, the numbering of the table is not Table 5 but Table S5). It should be cited as table S5. This error must be corrected so as not to hinder the reading of the article.
Response: Thanks. We have corrected all cited tables ad figures in articles as Table S or Figure S.
Please rewrite the phrase "Except difference method was done using SAS 9.4 version," as it can confuse. I suggest the following wording:
SAS 9.4 was used for performing the exact difference method. Other analyses were done using R software, version 4.2.1.
Response: Thanks. We have rewritten the phrase as “SAS 9.4 was used for performing the exact difference method. Other analyses were done using R software, version 4.2.1”.
Please review lines 287-288. Not understood "Healthy behaviours as important influencing factors for health," It should probably be "are" Instead of "as"
Response: Thanks. We have corrected this sentence “Healthy behaviours as important influencing factors for health, previous studies reported that it might alleviate the risk of deaths” as “A healthy lifestyle is an important influencing factor for health, and previous studies have reported that it may also alleviate the risk of death”. We have corrected language issues of this article. Besides, all the manuscript has been copy edited for proper English language at LetPub which is an author service brand owned and operated by Accdon LLC. The language editors are native English speakers with long-term experience in editing scientific and technical manuscripts.
Round 2
Reviewer 1 Report
thanks to the authors for their corrections and additions, I think this article is publishable in this form.